# Association between Maternal Factors, Preterm Birth, and Low Birth Weight of Chilean Singletons

**DOI:** 10.3390/children9070967

**Published:** 2022-06-28

**Authors:** Alejandra Rodríguez-Fernández, Marcela Ruíz-De la Fuente, Ximena Sanhueza-Riquelme, Julio Parra-Flores, María Dolores Marrodán, Eduard Maury-Sintjago

**Affiliations:** 1Department of Nutrition and Public Health, Universidad del Bío-Bío, Chillan 3780000, Chile; alrodriguez@ubiobio.cl (A.R.-F.); marcelaruiz@ubiobio.cl (M.R.-D.l.F.); xsanhue@ubiobio.cl (X.S.-R.); juparra@ubiobio.cl (J.P.-F.); 2GABO—Grupo de Investigación en Auxología, Bioantropología y Ontogenia, FACSA, Universidad del Bío-Bío, Chillán 3780000, Chile; 3Escuela de Nutrición y Dietética, Universidad del Bío-Bío, Chillan 3780000, Chile; 4Departamento de Biodiversidad, Ecología y Evolución, Grupo de Investigación EPINUT (ref. 920325), Universidad Complutense de Madrid, 28040 Madrid, Spain; marrodan@bio.ucm.es

**Keywords:** preterm birth, low birth weight, maternal factors, newborn

## Abstract

There has been an increase in preterm (PT) births in Western countries in recent years, which is associated with low-birthweight (LBW) children. The aim of this study was to determine the association between maternal factors and PT and LBW Chilean newborns. Methods: This was an analytical cross-sectional study of a national sample of 903,847 newborns and their mothers. The newborn gestational age, birth weight, maternal age, marital status, education, employment situation, and residence were analyzed. A multivariate logistic regression model was applied (α = 0.05) (STATA v.15). The prevalence was 6.8% and 5.0% for PT and LBW, respectively. The probability of the newborns being PT and LBW was 1.18 and 1.22 times if their mothers had <12 years of education and 1.38 and 1.29 times if the mothers were ≥35 years old, respectively. Mothers with <12 years education and ≥35 years were risk factors for PT and LBW newborns. Maternal educational attainment was a protective factor for the Chilean newborns, and a maternal age ≥35 years was a risk factor for PT and LBW.

## 1. Introduction

The increased number of preterm (PT) newborns in recent decades in the Western Hemisphere is considered a worldwide epidemic because of its high incidence and associated risk [1]. The United States is the Western country with the highest rate of PT births, with a reported increase from 9.4% (1980) to 10.1% (2018); the same trend has been identified in other Western countries, such as Austria, Germany, Canada, and Chile [2,3,4]. Although term newborns can have low birth weight (LBW), the risk increases in preterm newborns. Preterm birth and LBW are conditions that increase the risk of morbidity and mortality in newborns [5].

The etiological causes of PT birth are numerous, including multiple pregnancies, maternal, fetal, and placental abnormalities, substance abuse, ethnic characteristics, and maternal and social factors [6]. Fuchs et al. [7] showed that Canadian mothers aged > 40 years were at higher risk (OR 1.20; 95% CI 1.06–1.36) of having PT children. Temu et al. [8] studied Tanzanian women and found that the risk of PT pregnancy increased when the mother lived alone (OR 5.26; 95% CI 1.11–25.14) or had not completed formal education (OR 1.20; 95% CI 1.06–3.55).

Few studies in Chile have evaluated the maternal factors associated with PT birth, and these have presented some methodological problems and inconsistent results. Araya et al. [9] studied Chilean newborns between 1994 and 2013; they indicated that both educational attainment and maternal age were risk factors for having PT babies. However, their database included both single and multiple pregnancies, which constitutes bias because the frequency of PT birth in multiple pregnancies is approximately 60% [2,10].

López-Orellana [11] studied Chilean data, which showed that there was an increase in the number of mothers > 35 years of age between 1991 (10.6%) and 2012 (16.7%). The increase in maternal age was also reported by other authors, such as Qin et al. [12] and Cooke and Davidge [13]. Chilean studies using updated national databases are limited or non-existent.

The aim of this study was to determine the association between PT birth and LBW and maternal factors in Chilean singles.

## 2. Materials and Methods

### 2.1. Design and Sample

This was an analytical cross-sectional study. Total live births in Chile from 2014 to 2017 were 959,165, and the sample consisted of 903,847. Data included in the study were taken from public records of the Department of Health Statistics and Information of the Chilean Ministry of Health (www.minsal.cl). The study excluded newborns from multiple births (n = 20,584), <24 weeks gestation (n = 1103), birth weight < 500 g (n = 677), and/or records in which some of the maternal data were missing (n = 32,954) (Appendix A: flowchart of inclusion and exclusion, Appendix A).

### 2.2. Study Variables

The following categories were established for analysis: delivery term (≥37 weeks or preterm (PT) <37 weeks), birth weight (normal ≥2500 or low <2500 g), maternal age (<35 years or ≥35 years), marital status (married/living with a partner or single), maternal educational attainment (<12 years or ≥12 years), working mother (yes or no), and residence (urban or rural). The neonatal nutritional evaluation used standardized references established for Chile and PT was defined according to the proposal by Quinn et al. [14].

### 2.3. Statistical Analysis

The statistical analysis included a description with absolute-frequency and percent-age measurements. Bivariable analyses were conducted using Pearson chi-squared tests. To evaluate association, a sex-adjusted multiple logistic regression model was implemented for the PT birth variable and maternal factors, while another model was applied for the LBW variable and maternal factors (logit (PT)= 2.532 + 0.323 MA + 0.165 ME − 0.081 L − 0.061 WM − 0.021 MS − 0.246 S; logit (LBW)= 2.970 + 0.259 MA + 0.204 ME − 0.093 L − 0.067 WM − 0.010 MS − 0.009 S). In both cases, the odds ratio (OR) and confidence interval (95% CI) were included. The Hosmer–Lemeshow test was used to evaluate model fit (for comparative purposes, a third model was performed with maternal age > 40 years (Appendix A: Equations for the PT and LBW model with maternal age > 40 years, Appendix A). Data processing was performed with STATA (v.15) software at α < 0.05 significance level.

## 3. Results

The newborn distribution by sex was 51% female and 49% male. In total, 6.8% of the newborns were PT and 5% had LBW. For the mothers, 18.1% were <35 years old and 50.7% were single. The level of educational attainment was predominantly >12 years (90.0%), 50.3% of the mothers were not working outside the home, and 91.5% lived in urban areas (Table 1). In addition, there was a statistically significant increase in the percentage of PT delivery, LBW, and mothers who were ≥35 years old (*p* < 0.001) (Appendix A: annual increase in the prevalence of prematurity, low birth weight, and maternal age, Appendix A).

Table 2 shows a significant relationship between PT and LBW with the variables evaluated (*p* < 0.05). Only marital status was not related to LBW (*p* = 0.232). A detailed analysis of the characteristics of the newborns is available in Appendix A (Appendix A: characteristics of newborns).

The logistic regression model showed that both a maternal age ≥35 years (OR 1.382; 95% CI 1.355–1.410) and educational attainment <12 years (OR 1.180; 95% CI 1.148–1.213) represented risk for the PT birth variable. In addition, urban living, employment, and having a partner were potentially protective variables for the PT newborns (Table 3).

In the case of LBW, the model indicated that maternal age ≥35 years and educational attainment <12 years increased the risk by 1.296 and 1.224 times, respectively. The maternal variables of urban living and employment protected against low weight. The marital-status variable was not significant (Table 3).

## 4. Discussion

The maternal factors relating to educational attainment and maternal age ≥ 35 years are highlighted as important risk factors for the evolution of pregnancy and the perinatal period, regardless of the income of the mother’s country of residence [15].

When comparing our results with those published by López-Orellana [11], there is evidence that the numbers of PT newborns (2012: 6.0% vs. 2017: 6.8%) and mothers ≥ 35 years old have increased rapidly in recent years (2012: 16.7% vs. 2017: 19.6%); these findings concur with the global trend [16]. The postponement of motherhood has social origins associated with women joining the work force, the contraceptive revolution, women’s liberation, and advances in assisted reproductive technology (ART) [17,18].

Only 9.1% of the studied mothers had an educational attainment of <12 years, while national statistics show that 25.6% of the total population have <12 years of educational attainment, and the mean educational attainment for women is 10.95 years [19]. We also found that mothers with <12 years of education had a higher probability of having PT delivery (OR 1.180; 95% CI 1.148–1.213) and LBW (OR 1.224; 95% CI 1.187–1.263) children compared with those with higher levels of educational attainment. It was found that the high educational attainment of parents acts as a protective factor for LBW newborns, which was supported by our research [20]. Other studies have associated low educational attainment in mothers with a higher probability of LBW in children, and mothers with incomplete secondary education had 18% more risk of having LBW children [21,22].

In total, 18.1% of the women were ≥35 years old. It has been shown that a maternal age > 35 years is statistically related to PT birth (OR 1.45; 95% CI 1.38–1.53) and LBW (OR 1.37; 95% CI 1.26–2.50) newborns; the risk of fetal death also increases as maternal age increases [7]. These association values are close to those encountered in our study for PT birth (OR 1.382; 95% CI 1.355–1.410) and LBW (OR 1.296; 95% CI 1.266–1.327) newborn. It is therefore necessary to promote education for future fathers and mothers about the health risks involved in the mother–child relationship if motherhood is delayed after the age of 35, and more so after age 40 [23]. We also detected that maternal age > 40 years increases the risk of PT birth even further (OR 1.578; 95% CI 1.524–1.634) and LBW (OR 1.587; 95% CI 1.526–1.651) (Appendix A: Multiple logistic regression model for the association between preterm birth, low birth weight, and maternal factors (maternal age > 40 y.o.), Appendix A).

Our findings show that mothers who reside in urban areas have a lower risk of PT births (OR 0.92; 95% CI 0.89–0.95) and LBW (OR 0.91; 95% CI 0.88–0.94). Perez-Patron et al. [24] studied single-mother PT births in the United States and showed that urban areas (7.9%) have a lower rate of PT births than rural areas (8.3%). This concurs with the findings of Baer et al. [25], who reported a lower risk of PT deliveries in urban areas (RR 0.8; 95% CI 0.7–0.9). However, studies that evaluate LBW based on residence show contradictory results. Kaur et al. [26] indicated a lower prevalence of LBW in urban areas (2.0%) than in rural areas (9.8%). Meanwhile, Rodríguez-López et al. [27] described a higher prevalence of LBW that was proportional to the size of the city (PRR 1.06; 95% CI 1.01–1.12) in a study in eight Latin American countries. The location of residence is important as a risk factor for PT birth and LBW; it has even been reported to influence neonatal mortality [28].

Maternal employment is a protective factor for both PT birth (OR 0.94; 95% CI 0.92–0.95) and LBW (OR 0.93; 95% CI 0.91–0.95). These findings contradict those reported by Saurel-Cubizolles et al. [29] and Agbla et al. [30], but they concur with the results shown by Rodrigues and Barros [31], who indicated that unemployed mothers had a higher risk of preterm delivery (OR 1.5; 95% CI 1.18–1.88). As with PT birth, maternal employment shows different results for LBW. Girma et al. [32] and Mahmoodi et al. [33] mentioned that employed women have a higher prevalence and risk of having LBW children. Mahmoodi et al. [33] indicated that this association is mainly due to unfavorable working conditions, such as humid environments, the use of detergents, and standing for many hours at a time. Information on workplace characteristics was unavailable when conducting the present study. This was a limitation, because differentiated risks have been reported in relation to types of work and the duration of the workday [34].

The mothers who were married or living with a partner showed a lower risk of PT birth (OR 0.97; 95% CI 0.96–0.99) than the single mothers. This concurs with the results reported by Zeitlin et al. [35] for European mothers (OR 1.61; 95% CI 1.26–2.07) and by Hidalgo-Lopezosa [21] for mothers in Spain (OR 1.14; 95% CI 1.11–1.18). The fact that this factor was not associated with LBW supported the findings of Girma et al. [32]. However, this differs from the results presented by Farbu et al. [36], who showed that single mothers living alone were indeed at risk (OR 1.37; 95% CI 1.13–1.66), while no association was found for when the mothers lived with their parents (OR 0.98; 95% CI 0.63–1.53). Since information on the characteristics of the single mothers was not available when the present study was conducted, it was not possible to investigate these associations in greater detail.

The present study has many strengths because it considered a large number of newborns that reflect national data. However, it has some limitations that should be addressed to ensure an appropriate interpretation of its results, including the fact that the database had no information regarding maternal or perinatal morbidity, which could have contributed to a more in-depth analysis of our findings.

## 5. Conclusions

The children of mothers with <12 years educational attainment have a higher probability of being preterm and having low birth weight. Furthermore, with mothers aged ≥35 years, this probability increases. Therefore, maternal educational attainment is a protective factor, and a maternal age of ≥35 years is a risk factor.

## Figures and Tables

**Table 1 children-09-00967-t001:** Studied maternal factors of Chilean newborns.

Variable	*n* = 903,847	%
Delivery		
Term (≥37 weeks)	841,983	93.2
Preterm (<37 weeks)	61,864	6.8
Birth weight		
Normal (≥2500 g)	858,434	95.0
Low (<2500 g)	45,413	5.0
Maternal age		
<35 years	739,966	81.9
≥35 years	163,881	18.1
Marital status		
Married or living with a partner	445,182	49.3
Single	458,665	50.7
Maternal educational attainment		
<12 years	82,663	9.1
≥12 years	821,184	90.9
Working mother		
Yes	448,984	49.7
No	454,863	50.3
Residence		
Urban	826,628	91.5
Rural	77,219	8.5

**Table 2 children-09-00967-t002:** Bivariate analysis of factors associated with preterm and low-birth-weight newborns in Chile.

Variable	Preterm Birth			Low Weight Birth		
	Number	Yes	*p*	Number	Yes	*p*
Sex						
Female	426,007 (92.4)	35,067 (7.6)	<0.001	437,013 (94.8)	24,061 (5.2)	<0.001
Male	415,976 (94.0)	25,797 (6.0)		420,421 (94.9)	22,352 (5.1)	
Age						
<35 years	692,262 (93.6)	47,704 (6.5)	<0.001	704,522 (95.2)	35,444 (4.8)	<0.001
≥35 years	149,721 (91.4)	14,160 (8.6)		153,912 (93.9)	9969 (6.1)	
Marital status						
Married or living with a partner	414,370 (93.1)	30,812 (6.9)	0.004	422,690 (94.9)	22,492 (5.1)	0.232
Single	427,613 (93.2)	31,052 (6.8)		435,744 (95.0)	22,921 (5.0)	
Maternal educational attainment						
<12 years	765,971 (93.3)	51,213 (6.7)	<0.001	780,823 (95.1)	40,361 (4.9)	<0.001
≥12 years	76,012 (92.0)	6651 (8.0)		77,611 (93.9)	5052 (6.1)	
Working mother						
Yes	423,164 (93.0)	31,699 (7.0)	<0.001	431,369 (94.8)	23,494 (5.2)	<0.001
No	418,819 (93.3)	30,165 (6.7)		427,065 (95.1)	21,919 (4.9)	
Residence						
Urban	769,802 (93.1)	56,826 (6.9)	<0.001	784,894 (94.9)	41,734 (5.1)	0.001
Rural	72,181 (93.5)	5038 (6.5)		73,540 (95.2)	3679 (4.8)	

**Table 3 children-09-00967-t003:** Multiple logistic regression model for the association between preterm birth, low birth weight, and maternal factors.

Variable	Preterm Birth		Low Birth Weight	
	OR_crude (95% CI)_	OR_adjusted (95% CI)_	OR_crude (95% CI)_	OR_adjusted (95% CI)_
Maternal age (≥35 years)	1.372 (1.346–1.401)	1.382 (1.355–1.410)	1.287 (1.258–1.317)	1.296 (1.266–1.327)
Educational attainment (<12 years)	1.214 (1.182–1.246)	1.180 (1.148–1.213)	1.259 (1.222–1.298)	1.224 (1.187–1.263)
Residence (urban)	0.946 (0.918–0.974)	0.922 (0.895–0.950)	0.941 (0.909–0.974)	0.911 (0.880–0.943)
Working mother	0.961 (0.946–0.977)	0.941 (0.925–0.958)	0.942 (0.925–0.960)	0.935 (0.917–0.954)
Marital status (married or living with a partner)	0.977 (0.961–0.993)	0.979 (0.963–0.995)	0.989 (0.970–1.007)	0.990 (0.971–1.009)

Adjusted for sex.

## Data Availability

Please check www.minsal.cl for Chilean newborn data.

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
