# Peer review of "Association between Maternal Factors, Preterm Birth, and Low Birth Weight of Chilean Singletons"

_children, 2022, doi:10.3390/children9070967_

Round 1
Reviewer 1 Report
The authors have presented a population based analysis of factors related to preterm birth and low birth weight for the country of Chile. The paper is interesting. The approach is sound and the conclusions reached are justifiable.
I have several comments and suggestions to make:
The study only deals with singleton pregnancies. I think that this should be included in the title, given the known strong association between multiple pregnancy and preterm birth.
The size of the study is a little unclear. Is 903,847 the total number of the whole birth cohort? Or the number included in the analysis? If this is the birth cohort, the authors should include the size of the whole analysed cohort also. If it is the analysed cohort, the total size of the birth cohort would be of interest. It would also be of interest to have information on the number of babies not included in the analysis (multiples, <24 weeks, <500g and those with missing data).
It is not clear how the authors chose the variables that they included in their multiple logistic regression models. For example, why did they use 35 years as an age cut off? It is not possible to say, from this analysis, at what age the risks increase.
Why did they use 12 years education as the cut off? Is that a reflection of the Chilean Education System?
Page 79 “Some 6.8% of newborns 79 were preterm and 5% had low birth weight.” There is clearly overlap between these 2 variables, which has not been discussed at all in this paper. Some indication of the overlap, possibly using a Venn diagram would be of interest.
It is really important to consider the fact that Preterm Birth and Low Birth Weight are co-related and to consider this during the analysis and discussion. For example, do the identified risk factors that they have seen to increase LBW (education status and older age) increase LBW only because of the increase in PT? or do these factors affect the prevalence of LBW independently of their impact on PT rate?
It may have been helpful to do a multiple regression analysis using Birth Weight and Gestation as continuous variables together.
Some data appear in the Discussion and not in the results section, which doesn’t make sense:
Discussion paragraph 2 – the state that the rate of PT has increased in their population during the observation period, but these data are not shown in the results section. Similarly they state that the proportion of mothers >35 years increased during the study period. These are interesting data and should have been presented more clearly.
Minor points:
Line 109 – “Only 9.1% of the studied mothers had an educational attainment of <12 years; this 109 value is lower than the national statistics in which the mean educational attainment for 110 women is 10.95 years” what is the national statistic for educational attainment <12 years?
Why did they adjust the MLR for sex? This should be explained.
Was the maternal “age” taken as the age at the time of birth or the time of conception?
Typographic error on Line 33 “pregnan-cies” should be “pregnancies”.
Reviewer 2 Report
Reviewer comments:
This is a good first draft of a manuscript of public health importance and in-line with the journal’s scientific mission. However, it needs a bit more work.
In general, there doesn’t seem to be enough information about the study included in the manuscript. I suggest the authors use the Strobe checklist to ensure all relevant details are in the paper. For instance, the Table 1 should be a comprehensive look at the demographic, medical, obstetric factors to help the reader under the underlying population. Not just the variables included in the regression model. Also why were the covariates categorized to be binary? Did the authors look at finer categorizations? Following the Strone guidelines will help the authors draft a stronger manuscript.
Did you consider looking at moderately and very early preterm as an outcome as well? Same for moderately and very LBW. What about post-term and high birthweight?
Why model PTB and LBW? Why not also include AGA, SGA and LGA?
Were stillborns included in this analysis or only livebirths?
Could you include early neonatal mortality? Or even day 0 or day 1 mortality?
The methods sections needs a short description of the MOH data. How complete is it? Is this data representative of the national population? Was it sampled? Or is this the whole dataset (minus the exclusions mentioned in lines 59-61). What proportion of babies were excluded due to each of the missing covariates and combined?
Do you have any other information in the data – like preconception maternal weight or maternal co-morbidities? This is where a description of the dataset and how it was collected would be helpful.
There are also numerous typos and the manuscript needs a good copy-edit and proofread. I haven’t noted them here.
Minor comments:
Line 14: Accompanied by an increase LBW?
Line 28: Could you also include prevalence estimates and how they have changed over time here?
Line 30: To investigate this, could you also look different combination of Term and LBW status? (example: Term+LBW, Preterm+LBW, etc). Or include size-for-gestational age as an outcome as mentioned above.
Line 44: Sentence needs a citation
Line 50: What is meant by “updated” databases? What databases did the Orellana, Qin, Cooke/Davidge studies use?
Lines 72 – 74: Please present the equations for the two models.
Line 126-128: Please include this in the Results section (and Methods section). Was this another model? This is also why looking at finer categories of the covariates is helpful.
Discussion: I had some questions/issues with the Discussion section but it would be helpful to have further details (using the Strobe checklist) before commenting. Please also include the c-section rate if available.
